# Ultrastructural and Photosynthetic Responses of Pod Walls in Alfalfa to Drought Stress

**DOI:** 10.3390/ijms21124457

**Published:** 2020-06-23

**Authors:** Hui Wang, Qingping Zhou, Peisheng Mao

**Affiliations:** 1Forage Seed Laboratory, Key Laboratory of Pratacultural Science, Beijing Municipality, China Agricultural University, Beijing 100193, China; huiwang@swun.edu.cn; 2College of Qinghai-Tibetan Plateau, Southwest Minzu University, Chengdu 610041, China; qpingzh@aliyun.com

**Keywords:** pod wall, nonleaf green organs, ultrastructure, proteomic, alfalfa

## Abstract

Increasing photosynthetic ability as a whole is essential for acquiring higher crop yields. Nonleaf green organs (NLGOs) make important contributions to photosynthate formation, especially under stress conditions. However, there is little information on the pod wall in legume forage related to seed development and yield. This experiment is designed for alfalfa (*Medicago sativa*) under drought stress to explore the photosynthetic responses of pod walls after 5, 10, 15, and 20 days of pollination (DAP5, DAP10, DAP15, and DAP20) based on ultrastructural, physiological and proteomic analyses. Stomata were evidently observed on the outer epidermis of the pod wall. Chloroplasts had intact structures arranged alongside the cell wall, which on DAP5 were already capable of producing photosynthate. The pod wall at the late stage (DAP20) still had photosynthetic ability under well-watered (WW) treatments, while under water-stress (WS), the structure of the chloroplast membrane was damaged and the grana lamella of thylakoids were blurry. The chlorophyll a and chlorophyll b concentrations both decreased with the development of pod walls, and drought stress impeded the synthesis of photosynthetic pigments. Although the activity of ribulose-1,5-bisphosphate carboxylase (RuBisCo) decreased in the pod wall under drought stress, the activity of phosphoenolpyruvate carboxylase (PEPC) increased higher than that of RuBisCo. The proteomic analysis showed that the absorption of light is limited due to the suppression of the synthesis of chlorophyll a/b binding proteins by drought stress. Moreover, proteins involved in photosystem I and photosystem II were downregulated under WW compared with WS. Although the expression of some proteins participating in the regeneration period of RuBisCo was suppressed in the pod wall subjected to drought stress, the synthesis of PEPC was induced. In addition, some proteins, which were involved in the reduction period of RuBisCo, carbohydrate metabolism, and energy metabolism, and related to resistance, including chitinase, heat shock protein 81-2 (Hsp81-2), and lipoxygenases (LOXs), were highly expressed for the protective response to drought stress. It could be suggested that the pod wall in alfalfa is capable of operating photosynthesis and reducing the photosynthetic loss from drought stress through the promotion of the C4 pathway, ATP synthesis, and resistance ability.

## 1. Introduction

Photosynthesis is considered as the most important chemical reaction and provides over 90% of dry matter for crop yield formation [1,2]. Increasing crop yield by promoting photosynthesis has been the research hotspot until now [3]. Green leaves are commonly focused as the main source for producing photosynthate. However, nonleaf green organs (NLGOs) have been proven to be practically or potentially capable of assimilating CO_2_. Many scientists have previously reported that the silique shell of oil rape (*Brassica napus*) [4]; the boll shell of castor (*Ricinus communis*) [5]; the pod wall of legume crops, including chickpea (*Cicer arietinum*) [6], soybean (*Glycine max*) [7], and alfalfa (*Medicago sativa*) [8]; ears of cereal including rice (*Oryza sativa*) [9], barley (*Hordeum vulgare*) [10], and wheat (*Triticum turgidum*) [11]; flowers [12], stems [13], and roots [14] in some plants could photosynthesize and make an important contribution to yield formation. In addition, under drought conditions, the photosynthetic contribution of NLGOs turn greater, and NLGOs even become the primary photosynthetic organs for grain-filling [9,15].

Photosynthesis in leaves is sensitive to water deficit. Under water-deficit conditions, the flag leaves of cereal crops will wilt and senesce, while NLGOs are able to maintain relatively high water content due to some special features, including xeromorphic anatomy [9], lower stomatal conductance and transpiration rate [16], and a higher ability for osmotic adjustment [17]. In addition, NLGOs show better photosynthetic performance in the change of stomatal densities [18], chlorophyll concentration [5,19], and photosynthetic enzyme abilities [20] than leaves in response to water deficit. NLGOs have been proved to play an important role in regulating carbon partitioning during grain-filling to compensate for the reduction due to the decrease of photosynthetic ability in leaves [7]. Shreds of evidence based on anatomical, physiological, and molecular research have shown that the high photosynthetic efficiency pathway, C4-like or C3-C4 intermediate photosynthesis, might exist in the NLGOs of C3 crops [11]. Kranz anatomy is considered a crucial characteristic in C4 crops. The ear organs, including glume, lemma, and awn in C3 cereals, have two types of chloroplasts existing, respectively, in two types of cells, mesophyll cells and the cells arranged around the vascular bundles, similar to maize (*Zea mays*) leaves [11]. One of the key photosynthetic enzymes, phosphoenolpyruvate carboxylase (PEPC), has been detected with activity in NLGOs, and PEPC has a higher ability than ribulose-1,5-bisphosphate carboxylase (RuBisCo) under drought stress [21,22,23]. Besides PEPC, other enzymes, including NAD-dependent malic enzyme (NAD-ME), NADP-dependent malic enzyme (NADP-ME), and NADP-dependent malate dehydrogenase (NADP-MDH) involved in the C4 photosynthetic cycle, have been induced in NLGOs under drought stress [21]. Some genes, including *ppc*, *aat*, *mdh*, *me2*, *gpt*, and *ppdk*, specific to NAD-ME type-C4 photosynthesis, have been identified in wheat caryopsis [24]. However, other scientists have proposed the negative hypothesis that C4 photosynthesis is lacking in C3 crops. Singal et al. reported that CO_2_ was assimilated by C4 photosynthesis in the pericarp of wheat, but not in awn and glume [25]. A C4 photosynthesis metabolism occurring in C3 crops depends on ontogeny differences, cultivars, and environments like high or low CO_2_ and heat or drought stress [24,26]. Above all, the response mechanism of carbon fixation in NLGOs to drought stress is still unclear.

Alfalfa is widely cultured around the world to produce high-quality hay for feeding livestock, especially dairy cows. Seed producers have long focused on alfalfa seed yield increase. Moderate drought contributes to achieving higher seed yield during the flowering and seed maturation period. Nevertheless, little is known on the physiological response and the photosynthetic contribution of the pod wall in alfalfa under drought stress. Investigating and increasing the photosynthetic ability of NLGOs, especially under stress conditions, is a novel way to increase the photosynthetic ability of the whole plant and finally increase the grain yield. In this study, physiological, ultrastructural, and proteomic analyses were carried out to (1) investigate the photosynthetic characteristics of the pod wall in alfalfa, and (2) research the response mechanism of photosynthesis in the pod wall to drought stress. 

## 2. Results

### 2.1. Changes of the Surface Characteristics and Ultrastructure of Pod Wall under Drought Stress

Stomata and epidermal hair were distinctly observed in the outer surface of the pod wall (Figure 1A,B). Stomata were composed of two semilunar guard cells encircled by several subsidiary cells. Stomata were open under both WW and WS, and the thick inner wall of stomata, the bright color part, could be clearly observed. In addition, a hump occurred around the base of epidermal hair, and lots of dots existed on the epidermal hair. Cells of the inner surface of the pod wall were tightly arranged together (Figure 1C).

Under WW, chloroplasts in the pod wall had the ability to photosynthesize from DAP5 to DAP20. Chloroplasts on DAP5 existed with the intact structure and were arranged close to the cell wall (Figure 2A). Chloroplast membrane structure was intact, and grana lamella was arrayed along the long axis of the chloroplast, some of which had already produced starch grains. More and bigger starch grains were produced in the chloroplasts on DAP10 (Figure 2B) and DAP15 (Figure 2C). The pod wall on DAP20 still had photosynthetic activity, while the cells had started to age and the nuclei were degrading. Few osmiophilic granules were found in cells (Figure 2D).

Under WS, chloroplasts were able to produce photosynthate on DAP5 and DAP10, while the structure of chloroplasts was gradually damaged from DAP15 to DAP20. The chloroplasts had intact membrane structures and had already started to produce starch grains on DAP5 (Figure 2E), and they produced more and bigger starch grains on DAP10 (Figure 2F). Lots of starch grains could still be observed on DAP15, while the evident changes occurred in the structure of chloroplasts, i.e., the membrane was partly broken, and the grana lamellae of thylakoids became blurry (Figure 2G). Few starch grains existed on DAP20, while lots of osmiophilic granules were presented. The membrane of chloroplasts was seriously broken, and the structure of thylakoids was blurring (Figure 2H).

Except for chloroplasts, the structure of other organelle or tissues changed under drought stress as well. The central vacuole was bigger in the cell under WW (Figure 2D), while the gap between the central vacuole and the cell wall become wider under WS (Figure 2H). The membrane structure of the mitochondrion was intact and clear on DAP5 under WS (Figure 2E), while it was broken and blurred on DAP20 (Figure 2H).

### 2.2. Changes of Chlorophyll Concentration in Pod Wall under Drought Stress

With the development of the pod wall, the concentration of chlorophyll a, chlorophyll b, and total chlorophyll decreased under both WW and WS treatments, and the concentration of chlorophyll a was higher than that of chlorophyll b, respectively. Furthermore, the concentration of chlorophyll a, chlorophyll b, and total chlorophyll in the treatment of WS decreased significantly (*p* < 0.05) compared with WW (Table 1).

### 2.3. Changes of Photosynthetic Enzyme Activities in Pod Wall under Drought Stress

PEPC and RuBisCo activities in pod walls both present a declining trend with the pod development (Figure 3). However, there were different responses for PEPC and RuBisCo in the treatment of WS. As a comparison with WW, PEPC activities could be increased significantly (*p* < 0.05) in the treatment of WS, while RuBisCo activities were decreased.

### 2.4. Proteomic Analysis on the Response of Pod Wall to Drought Stress

According to the Medicago database, a total of 4215 proteins were identified in the samples of pod walls (Appendix A). Under the WW, 373 proteins were significantly upregulated (fold change > 2.0, *p* < 0.05) for WW15 vs. WW10, and 101 proteins were significantly downregulated for WW20 vs. WW10 (fold change < 0.5, *p* < 0.05, Figure 4). For the WS, there were 184 upregulated proteins obtained in WS20 vs. WS10, and the number of downregulated proteins was lowest in WS20 vs. WS15. In addition, 144 proteins were significantly downregulated in WS10 vs. WW10, while 48 proteins were significantly downregulated in WS15 vs. WW15. 

According to the Kyoto Encyclopedia of Genes and Genomes (KEGG) enrichment analysis, during pod development, most of the identified chlorophyll a–b binding proteins involved in the photosynthesis–antenna proteins were significantly downregulated in pod walls under WW or WS (Figure 5, Appendix A). The synthesis of chlorophyll a–b binding proteins in the pod wall was restricted by the drought stress on DAP10 (Figure 5, Appendix A).

Under the treatment of WS, some proteins, including ribulose-phosphate 3-epimerase, ctosolic fructose-1 6-bisphosphatase, fuctose-1, 6-bisphosphatase, fructose-bisphosphate aldolase, glyoxysomal malate dehydrogenase, malate dehydrogenase, malic enzyme, and sedoheptulose-1,7-bisphosphatase, which are involved in carbon fixation of photosynthetic organisms, were significantly upregulated among different growth stages after pollination (Figure 5, Appendix A). In contrast, some proteins, including 26S proteasome on-ATPase regulatory subunit 6, aspartate aminotransferase, and glutamate-glyoxylate aminotransferase, were significantly downregulated at different durations after pollination. Furthermore, under the treatment of WW, proteins such as fructose-1, 7-bisphosphatase, fructose-bisphosphate aldolase, malic enzyme, and PEPC were significantly upregulated at different durations after pollination, while proteins of cytosolic triosephosphate isomerase, glyceraldehyde-3-phosphate dehydrogenase, and phosphoglycerate kinase were significantly downregulated.

The synthesis of cytosolic fructose-1 6-bisphosphatase, fructose-1, 6-bisphosphatase, fructose-bisphosphate aldolase, and malic enzyme was inhibited by the drought stress on DAP10, and the synthesis of glyoxysomal malate dehydrogenase and ribose-5-phosphate isomerase A were both inhibited on DAP15 and DAP20, respectively (Figure 5, Appendix A). Nevertheless, aspartate aminotransferase on DAP15 and cytosolic triosephosphate isomerase and phosphoglycerate kinase on DAP20 were significantly induced by drought stress.

Proteins involved in photosynthesis, including cytochrome b6-f complex iron-sulfur subunit, F0F1 ATP synthase subunit gamma, oxygen-evolving complex/thylakoid lumenal 25.6 kDa protein, and photosystem II oxygen-evolving enhancer protein, were significantly upregulated in the pod wall at different durations after pollination under WS, while light-harvesting complex I chlorophyll a/b binding protein, oxygen-evolving enhancer protein, photosystem I P700 chlorophyll a apoprotein a2, and photosystem II D2 protein were significantly downregulated (Figure 5, Appendix A). Under WW, there were some significantly downregulated proteins, including cytochrome b559 subunit alpha, cytochrome b6-f complex iron-sulfur subunit, light-harvesting complex I chlorophyll a/b binding protein, oxygen-evolving enhancer protein, and photosystem II D2 protein at different durations after pollination.

For the treatment of WS, some proteins, including oxygen-evolving enhancer protein, photosystem II oxygen-evolving enhancer protein, photosystem I reaction center subunit II, and photosystem I reaction center subunit N in the pod wall on DAP10, were significantly upregulated (Figure 5, Appendix A). However, there were some proteins presenting downregulation in the pod wall on DAP15, such as cytochrome b559 subunit alpha, cytochrome b6-f complex iron-sulfur subunit, ATP synthase subunit gamma, oxygen-evolving complex/thylakoid lumenal 25.6 kDa protein, photosystem I P700 chlorophyll a apoprotein a2, and photosystem I reaction center subunit II.

Meanwhile, some proteins were expressed differently and significantly in the amino sugar and nucleotide sugar metabolism, the ascorbate and aldarate metabolism, the beta-alanine metabolism, the carbon metabolism, the starch and sucrose metabolism, the citrate cycle, the glycine, serine, and threonine metabolism, the linoleic acid metabolism, oxidative phosphorylation, phagosome, plant–pathogen interaction, proteasome, and the alpha-linolenic acid metabolism in pod walls under drought stress (Table 2).

## 3. Discussion

### 3.1. Observation of Surface and Ultrastructure in the Pod Wall

Stomata were distinctly observed on the outer surface of the pod wall (Figure 1). Stoma acted in respiration and transpiration and allowed CO_2_ to enter for operating photosynthesis as well. Previous research has shown that stoma was also found on other NLGOs, such as the exposed peduncles of wheat [22], the silique shell of oilseed rape [4], and the capsule wall of castor [5]. In addition, chloroplast, the important site for doing the light reaction of photosynthesis, was found in cells of the pod wall (Figure 2). At the early stage of pod development, the structure of chloroplasts in the pod wall is well-organized and intact, and the photosynthate, starch grains, are already observed in the chloroplasts (Figure 2A,E). Similar results were reported when observing the ultrastructure of the pod wall of pea [27] and chickpea [6]. Under WW, the pod wall at the late stage even could produce starch grains (Figure 2D). NLOG could maintain functional activity at the late stage when the photosynthetic activity of leaves declined [8,22,28]. However, drought stress could damage the structure of chloroplast membranes and thylakoids, and few starch grains were produced (Figure 2H). In addition, the epidermal hair, existing on the outer surface of the pod wall, likely acts to prevent damage from direct sunlight and protect against water loss.

### 3.2. Response of Chlorophyll Concentration and Photosynthetic Enzyme Activities

In the pod wall of alfalfa, chlorophyll a and chlorophyll b could be detected, and the concentration of chlorophyll a, chlorophyll b and total chlorophyll decreased with the development of pods (Table 1). Similarly, the concentration of chlorophyll a, chlorophyll b, and total chlorophyll in cotton (*Gossypium hirsutum*) leaves and NLGOs, including bracts, stems and boll shells, decreased with bolls developing, and the decreasing rate of the concentration in stems and boll shells was lower than that in leaves at the late stage of boll [28]. The chlorophyll biosynthesis was inhibited in the pod wall under drought stress (Table 1), while the content change was lower in NLOG than in leaves to maintain relatively high photosynthetic capacity [15,20,23]. Except for chlorophyll, photosynthetic enzymes are crucial for operating photosynthesis as well. The activities of key enzymes in the C3 and C4 cycles in the pod wall were determined in the present study and the activities of PEPC and RuBisCo both decreased with the development of pods (Figure 3). In addition, the activity of PEPC was higher than that of RuBisCo, and drought stress could induce the activity of PEPC (Figure 3). Previous studies reported that PEPC could make more contributions to photosynthesis than RuBisCo in NLGOs [22]. Although the activity of RuBisCo decreased under drought stress in NLGOs, the increasing activity of PEPC could, in part, compensate to ensure dry matter production [21,23].

### 3.3. The Differential Expression of Proteins under Drought Stress

#### 3.3.1. Photosynthesis-Antenna Proteins

Six proteins identified in the pod wall were chlorophyll a–b binding proteins, which are the apoproteins of the light-harvesting complex of photosystem Ⅱ that existed on the membrane of chloroplasts [29]. I3SZG9 (Lhca3) is the PSI inner antenna protein (LCHI); Lhcb 1, Lhcb 4, Lhcb 5, and Lhcb 6 belong to the PSII inner antenna proteins (LCHII). In the plants, 50% of chlorophyll associated with LCHII play important roles in the regulation of light energy distribution and photoprotective reaction. In this study, two types of antenna proteins identified in the pod wall were downregulated with the development of pod, and their synthesis was limited by the drought stress (Figure 5; Table 1). Similarly, the antenna proteins were downregulated in sugarcane (*Saccharum officinarum*) [30] and cucumber (*Cucumis sativus*) [31] under drought stress, which implied that drought stress impeded the synthesis of antenna proteins, suppressing the absorption of light. In addition, the downregulation of these proteins could decrease energy and substance consumption to promote the operation of other physiological activities for resisting drought stress [32].

#### 3.3.2. Photosynthesis

According to KEGG enrichment analysis for photosynthesis, the identified protein complex participates in the reactions taking place on the thylakoid membrane. Proteins, including PetC, PsbS, PsbQ, PsbO, and PsbC, were downregulated under WW (Figure 5, Appendix A), which means that the photosynthetic ability of the pod wall decreases as the pod developes. In addition, the PSII components, including PsbE, PsbP, and PsbO, were downregulated under drought stress (Figure 5, Appendix A). PsbE is cytochrome b559 subunit α, and the set of these three proteins, PsbP, PsbO, and PsbQ, was bound to the luminal surface of PSII, oxidizing water molecules to release O_2_ [33]. PsaD, PsaN, and PsaB are subunits of the PSI complex, and their synthesis is suppressed by the drought stress as well. F0F1 ATP synthase subunit γ (F0F1-ATPases), largely existing in chloroplasts, mitochondria, and cell nuclei, was downregulated in this study, which resulted in the reduction of ATP synthesis [34] and the decline of photosynthetic ability [35]. The downregulation of F0F1-ATPases by drought stress was also found in poplar (*Populus yunnanensis*) [36] and soybean [37]. Drought stress damages the electron transfer system on the thylakoid to reduce the photosynthetic ability of the pod wall through suppressing the synthesis of F0F1-ATPases, cytochrome b6-f complex, and the proteins involved in the photosystem [37].

#### 3.3.3. Carbon Fixation in Photosynthetic Organisms

Under WW, PEPC, the key enzyme in the C4 cycle, was significantly upregulated in the pod wall on DAP15, compared with that on DAP10 (Figure 5, Appendix A). Drought stress inhibited the synthesis of RuBisCo in leaves in alfalfa [38]. The induced proteins in the C4 cycle could compensate for the decrease of photosynthetic ability resulting from the inhibition of proteins in the C3 cycle under drought stress [39]. NLGOs of C3 plants might operate C3–C4 intermediate photosynthesis or C4-similar photosynthesis [11]. Two cell types (mesophyll and Kranz cells) were localized in the ears of wheat [11], in which the activity of PEPC was higher than in flag leaves, and the activity of PEPC was higher than that of RuBisCo [40]. In addition, compared with WW, another C4-pathway enzyme, aspartate aminotransferase, was significantly upregulated in the pod wall on DAP15 under WS (Figure 5, Appendix A). The upregulation of aspartate aminotransferase contributed to increasing the stress resistance of plants and maintaining high photosynthetic abilities under stress [41]. Some enzymes participating in the reduction period of the C3 cycle were upregulated under drought stress, while others involved in the regeneration period of RuBisCo were downregulated (Figure 5, Appendix A). Fructose-1, 6-bisphosphatase and fructose-bisphosphate aldolase were downregulated in pod walls on DAP10 under WS in comparison with WW, while triosephosphate isomerase and phosphoglycerate kinase were upregulated in pod walls on DAP20. Phosphoglycerate kinase, belonging to an upstream acting enzyme in the C3 cycle, had interaction with PEPC and aspartate aminotransferase in the C4 cycle under drought stress, which was found in maize [42]. Drought stress could induce the synthesis of triosephosphate isomerase in rice [43] and maize [44] to ensure the operation of photosynthesis.

#### 3.3.4. Carbohydrate Metabolism

Starch grains were observed to be filling the chloroplasts in the pod wall on DAP10 under WS (Figure 2), which meant that starch synthase was induced in pod walls on DAP10 by drought stress (Table 2). The recent research reported that total carbohydrate content decreased in the pod wall of soybean under drought stress, but not the starch content [7]. Enhance expression of starch synthase presented the protective response to drought stress in the pod wall.

Identified proteins, including pyruvate dehydrogenase (PDH) E1 beta subunit and ATP-citrate lyase/succinyl (ACL)-CoA ligase involved in the citrate cycle, were upregulated in pod walls on DAP15 under WS (Table 2). PDH produces chemical energy, and drought stress could promote the expression of relative encoding genes in rice [45]. PDH is one component of the pyruvate dehydrogenase complex (PDC) that can oxidize pyruvate into acetyl-CoA and NADH. In addition, the overexpression of the *ACL* gene could enhance drought resistance in tobacco (*Nicotiana tabacum*) [46]. Some other proteins involved in the citrate cycle, including methylenetetrahydrofolate reductase, E1 subunit-like 2-oxoglutarate dehydrogenase, aconitate hydratase, isocitrate dehydrogenase [NADP] and ATP-citrate lyase/succinyl-CoA ligase, were downregulated in pod walls on DAP20 under drought stress (Table 2). The downregulation of these proteins impedes the carbohydrate metabolism in the pod wall at the late growth stage. Previous research has shown that the stagnate of carbohydrate metabolism in the plants under drought stress could cause the accumulation of sugar [47], which contributes to improving the osmotic potential to enhance drought tolerance [38].

#### 3.3.5. Energy Metabolism

Identified proteins, including archaeal/vacuolar-type H+-ATPase subunit A, ATP synthase subunit beta, ATP synthase subunit alpha, and ATP synthase D chain, were upregulated in pod walls on DAP15 (Table 2). Budak et al. reported that ATP synthase subunit CF1 and ATP synthase subunit alpha had a higher expression level in wild wheat, with stronger drought resistance [48]. Under drought stress, the upregulation of these proteins involved in ATP synthesis could ensure energy metabolism in the plants for maintaining the operation of main physiological activities [49]. Photosynthesis is sensitive to drought stress, while respiration is not. The normal operation of respiration could provide the necessary energy for decreasing the damage from drought stress [7].

#### 3.3.6. Other Metabolism

Chitinase plays an important role in resistance to stress [50,51]. The activity of chitinase is low in the plant, but drought stress can induce the expression of the chitinase gene in wheat [50] and faba beans (*Vicia faba*) [51]. Similar results were found in the present study, where chitinase was upregulated in the pod wall on DAP15 under drought stress (Table 2). Heat shock protein 90 (Hsp90) complex regulated proteins fold and degrade and maintain stable plant cells [52]. Heat shock protein 81-2 (Hsp81-2) is a member of Hsp90 family and can be induced by drought stress in arabidopsis (*Arabidopsis thaliana*) [53]. In the present study, Hsp81-2 involved in plant–pathogen interaction was identified and upregulated in the pod wall on DAP10 and DAP15 under drought stress. Most of the lipoxygenases (LOXs) identified were upregulated in the pod wall on DAP10 (Table 2). LOX participates in many activities in the plants, and drought stress caused the rapid accumulation of LOX mRNA in barley [54]. The overexpression of *CaLOX1* enhanced the resistance to drought stress in arabidopsis [55].

In summary, the structure of chloroplast in the pod wall is damaged at the late stage of development under drought stress, but not at the early stage. The synthesis of some proteins involved in photosystem I, photosystem II, and the regeneration period of RuBisCo in the pod wall at the early stage and TCA cycle at the late stage are impeded under drought stress (Figure 6). Nevertheless, drought stress can induce the activity of PEPC and promote the synthesis of some proteins participating in the pathway of the C4 cycle and energy metabolism at the early stage and the reduction period of RuBisCo at the late stage.

This study provides the ultrastructural, physiological, and proteomic changes in alfalfa pod walls under drought stress. The results suggest that the pod wall shows the capability of conducting photosynthesis and regulating the C4 photosynthetic pathway, ATP synthesis, and resistance metabolism to ensure the operation of physiological reactions under drought stress.

## 4. Material and Method

### 4.1. Material

*M. sativa* cv. Zhongmu No. 2 seeds were sown in the plastic pots (weight 0.5 kg, height 30 cm, base diameter 19 cm, and top diameter 25 cm) in a greenhouse in October 2015. Each pot was filled with a mixture soil of vermiculite, peat, and black soil by 2:1:1 (total soil weight 3.5 kg, water content 23%). Four seedlings with a similar growth status were kept in each pot when the seedlings’ height was around 10 cm. They were equivalently and adequately watered every two days. All plants were cut till 10 cm, and then pots were moved out of and nearby the greenhouse, without any shelter, in April 2016. Every day, each pot was weighed with an electronic scale and watered to 7 kg. From 18 April (before visible bud stage), a drought stress treatment, denoted by WS, was started by watering each pot to 4.5 kg, while the control, denoted by WW, was still watered to 7 kg. Plants in WW and WS treatments were pollinated artificially every 5 days from 6 June till 21 June. After pollination, the flowers pollinated were marked with hang tags. On 26 June, the pod walls of marked pods with 3 replicates were collected on day 5 (DAP5), day 10 (DAP10), day 15 (DAP15), and day 20 (DAP20) after pollination under WW and WS treatments.

### 4.2. Surface and Ultrastructure Characteristics Observation for Pod Wall

The pod walls of DAP10 under WW and WS treatments were used as samples to observe the surface characteristics by using a scanning electron microscope (manufacture Hitachi S-570, city Japan). The pod walls of DAP5, DAP10, DAP15, and DAP20 under WW and WS treatments were used as samples for taking images with a transmission electron microscope (manufacture Hitachi H-7500, city Japan) to observe ultrastructure characteristics. Pretreatments of the pod walls for observation by using the scanning and transmission electron microscopes were conducted according to [56].

### 4.3. Chlorophyll Concentration of Pod Walls Measurement

The pod walls on DAP5, DAP10, DAP15, and DAP20 under WW and WS treatments were sampled and cut into filaments, respectively. Chlorophyll a and chlorophyll b concentrations in each sample with three repetitions were determined by soaking in extracting solution, filling in the 10 mL centrifuge tube. The extracting solution was the mixture of acetone and absolute ethyl alcohol by the volume rate of 2:1. Absorbancy of extracting solution at 663 and 645 nm was determined by using a ultraviolet spectrophotometer (manufacture UH5300, city Japan) to calculate the concentration of chlorophyll a and chlorophyll b. The gross chlorophyll concentration was the sum of chlorophyll a and chlorophyll b.

### 4.4. Photosynthetic Enzyme Activities Assays

The pod walls on DAP5, DAP10, DAP15, and DAP20 under WW and WS treatments were sampled with three repetitions. Each sample was homogenized in extracting solutions using a pestle and mortar in ice. The extracting solution contained 1 M Tris-H_2_SO_4_ (pH 7.8), 5% glycerol, 7 mM DTT, and 1 mM EDTA. The homogenate was filtered, and then the filtrate was centrifuged at 8000× *g* for 10 min at 4 °C. The supernatant was saved in an ice bath for subsequent enzyme assay. Photosynthetic enzyme activities were determined using an ultraviolet spectrophotometer (UH5300, Hitachi, Japan) at 340 nm. The reaction solution for phosphoenolpyruvate carboxylase (PEPC) contained 0.1 mol L^−1^ Tris-H_2_SO_4_ (pH 9.2), 0.1 M MgCl_2_, 100 mM NaHCO_3_, 40 mM PEP, 1 mg mL^−1^ NADH and MDH [57]. The reaction mixture for ribulose-1,5-bisphosphate carboxylase (RuBisCo) contained 100 mM Tris-HCl (pH 7.8), 160 U/mL CPK, 160 U mL^−1^ GAPDH, 50 mM ATP, 50 mM phosphocreatine, and 160 U mL^−1^ phosphoglyceric kinase [58].

### 4.5. Proteomic Analysis

The pod walls on DAP10, DAP15, and DAP20 under WW (denoted by WW10, WW15, and WW20, respectively) and WS (denoted by WS10, WS15, and WS20, respectively) treatments were collected and stored at −80 °C. Then, 0.05 g tissues of each sample with 3 repetitions were ground in liquid nitrogen before the addition of 200 μL plant total protein lysis buffer containing 20 mM Tris-HCl (pH7.5), 250 mM sucrose, 10 mM EGTA, 1% Triton X-100, protease inhibitor, and 1 M DTT. The mixture was incubated on ice for 20 min. Plant cell debris was removed via centrifugation at a speed of relative centrifugal force of 15,000× *g* for 15 min at 4 °C. The supernatant was collected, and the rest was centrifugated again, as above. The protein concentration was determined with a Bio-Rad Protein Assay kit based on the Bradford method, using BSA as a standard at a wavelength of 595 nm. All independent protein extractions were performed.

Briefly, 60 μg protein samples were reduced with 5 μL 1 M DTT for 1 h at 37 °C, alkylated with 20 μL 1 M iodoacetamide (IAA) for 1 h in the dark, and then digested with sequencing-grade modified trypsin (Promega) for 20 h at 37 °C. Digested peptides were separated with chromatography using an Easy-nLC1000 system (Thermo Scientific) autosampler. The peptide mixture was loaded on a self-made C18 trap column (C18 3 μm, 0.10 × 20 mm) in solution A (0.1% formic acid), then separated with a self-made Capillary C18 column (1.9 μm, 0.15 × 120 mm), with a gradient solution B (100% acetonitrile and 0.1% formic acid) at a flow rate of 600 nL/min. The gradient consisted of the following steps: 0–10% solution B for 16 min, 10–22% for 35 min, 22–30% for 20 min, then increasing to 95% solution B in 1 min and holding for 6 min. Separated peptides were examined in the Orbitrap Fusion mass spectrometer (Thermo Scientific, Waltham, MA, USA), with a Michrom captive spray nanoelectrospray ionization (NSI) source. Spectra were scanned over the m/z range 300–4000 Da at 120,000 resolution. An 18-s exclusion time and 32% normalization collision energy were set at the dynamic exclusion window.

### 4.6. Statistical Analysis

The significance of differences between mean values of physiological parameters, including the chlorophyll concentration and the enzyme activities under WW and WS, were analyzed using an LSD test by software SAS version 8.0.

RAW files of mass spectrometry were extracted using the MASCOT version 2.3.02 (Matrix Science, London, UK). Mass spectrometry data were searched, identified, and quantitatively analyzed using the software of Sequest HT and Proteome Discover 2.0 (Thermo Scientific). The database used in this study was uniprot-Medicago.fasta. Protein species with at least two unique peptides were selected for protein species quantitation, and the relative quantitative protein ratios between the two samples were calculated by comparing the average abundance values (three biological replicates). Protein species detected in only one material (A-line or B-line), with at least two replicates considered to be presence/absence protein species. Additionally, Student’s *t*-tests were performed to determine the significance of changes between samples. A fold-change of >2 and *p*-value < 0.05 in at least two replicates were used as the thresholds to define differently accumulated protein species.

## Figures and Tables

**Figure 1 ijms-21-04457-f001:**
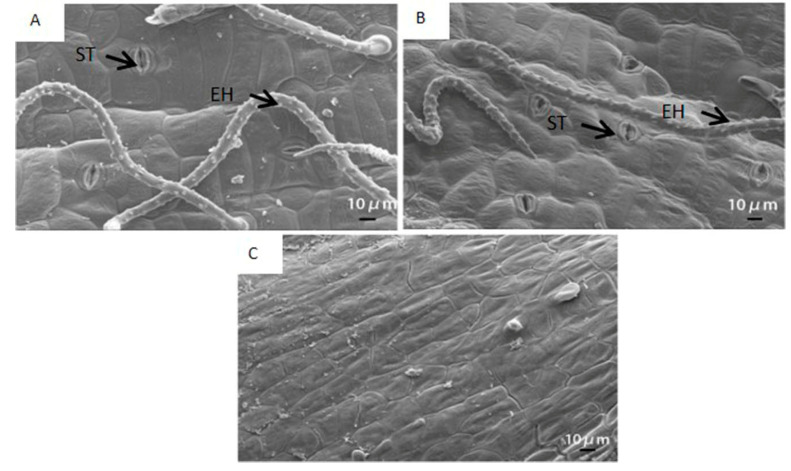
The scanning electron micrograph of the outer (**A**,**B**) and the inner surface (**C**) of the pod wall on the 10th day after pollination (DAP10) under well-watered (**A**) and water-stressed treatments (**B**). ST, stoma; EH, epidermal hair; D, dots; H, hump.

**Figure 2 ijms-21-04457-f002:**
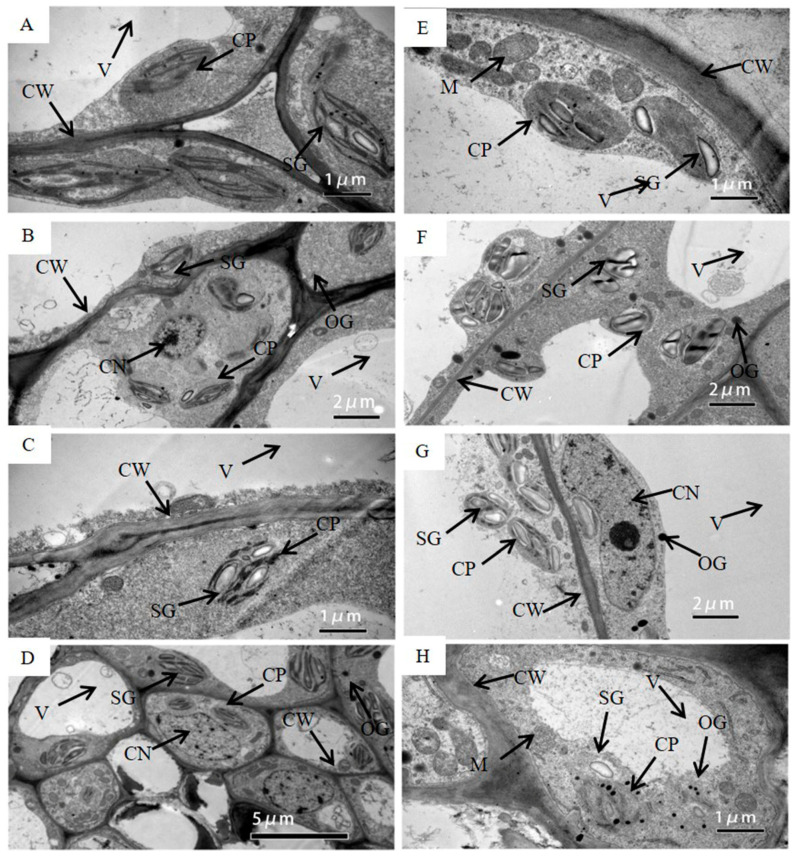
The transmission electron micrograph of cells in the pod wall on DAP5 (**A**,**E**), DAP10 (**B**,**F**), DAP15 (**C**,**G**), and DAP20 (**D**,**H**) under well-watered (**A**–**D**) and water-stressed treatment (**E**–**H**). CW, cell wall; CP, chloroplast; SG, starch grain; OG, osmiophilic granules; V, central vacuole; CN, cell nucleus; M, mitochondrion; T, thylakoid.

**Figure 3 ijms-21-04457-f003:**
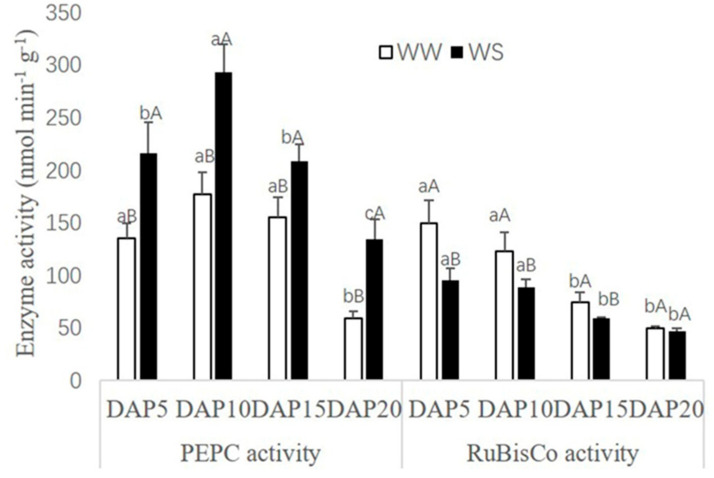
Effect of drought stress on photosynthetic enzyme activity (nmol min^−1^ g^−1^) in the pod wall. Different small letters up the white bar and different capitals up the black bar within one photosynthetic enzyme mean a significant difference at the 0.05 probability level. WW, well-watered; WS, water-stressed.

**Figure 4 ijms-21-04457-f004:**
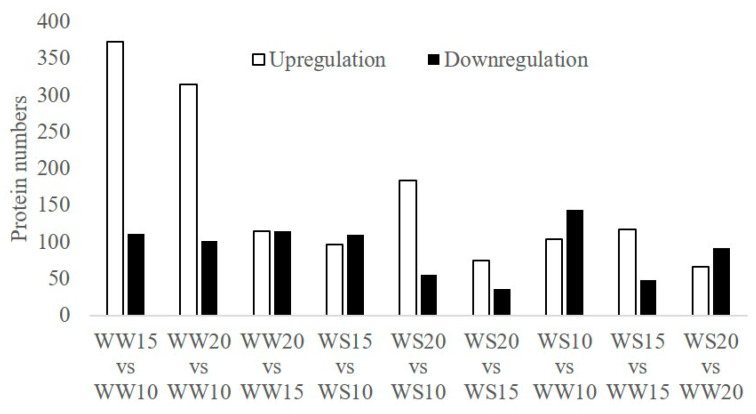
Number of proteins differently expressed in pod walls under drought stress. The white and black squares represent, respectively, significant upregulation and downregulation at the 0.05 probability level.

**Figure 5 ijms-21-04457-f005:**
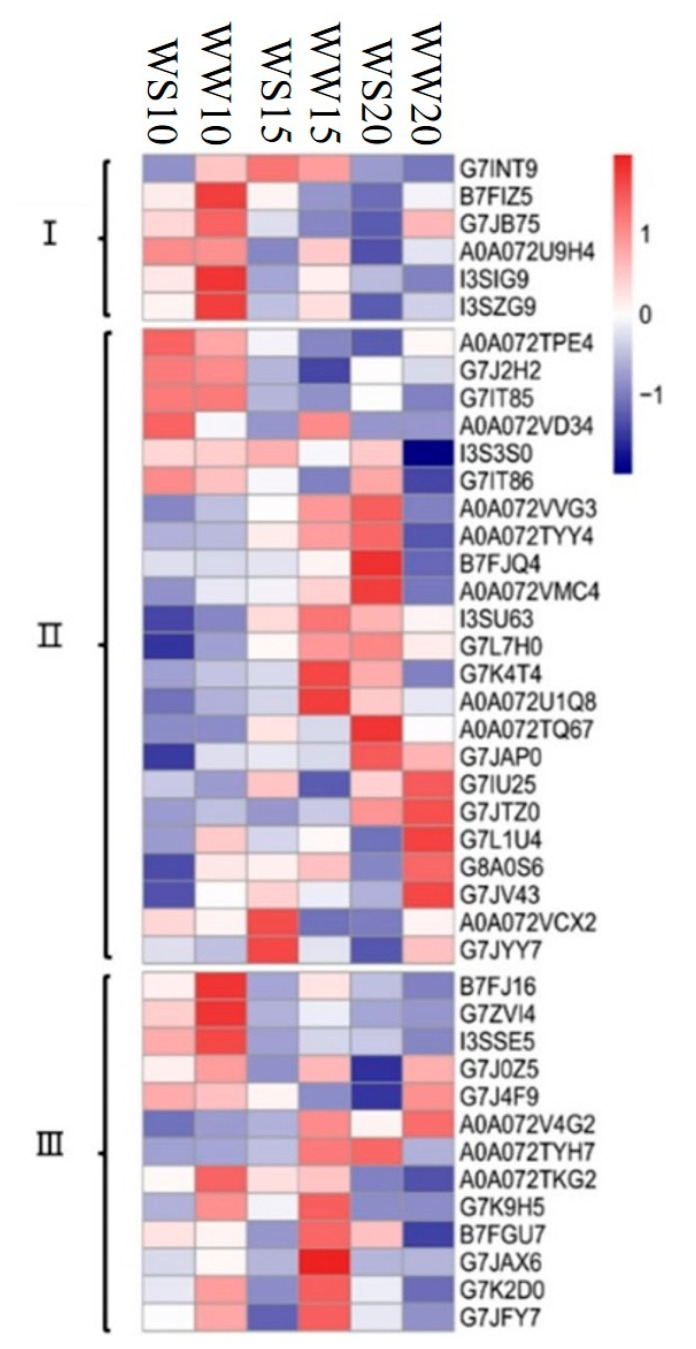
Cluster analysis of proteins involved in the significant pathway related to photosynthesis in pod walls at different durations after pollination under WW and WS. I, photosynthesis—antenna proteins; II, carbon fixation in photosynthetic organisms; III, photosynthesis.

**Figure 6 ijms-21-04457-f006:**
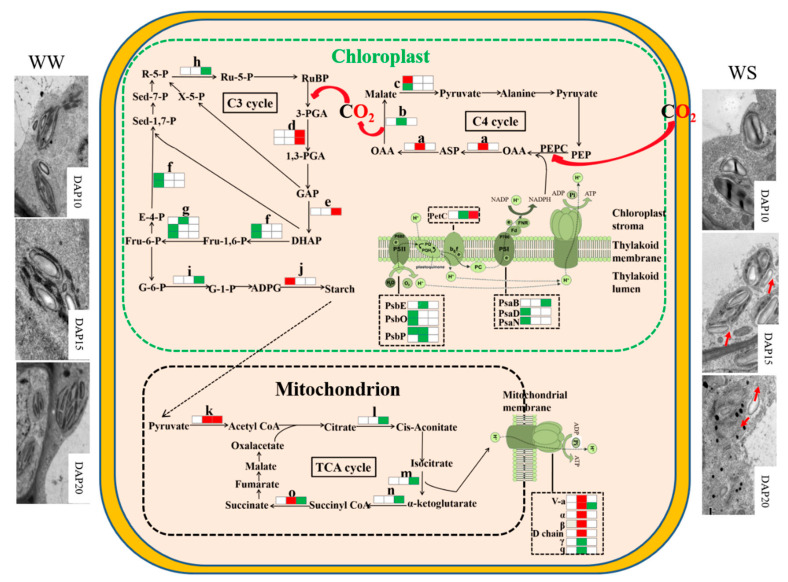
The pathways of proteomic mechanisms in the pod wall under drought stress. a, Aspartate aminotransferase; b, malate dehydrogenase; c, malic enzyme; d, phosphoglycerate kinase; e, triosephosphate isomerase; f, fructose-1,6-bisphosphate aldolase; g, fructose-1,6-bisphosphatase; h, ribose-5-phosphate isomerase; i, glucose-6-phosphate isomerase; j, starch synthase; k, pyruvate dehydrogenase E1 beta subunit; l, aconitate hydratase; m, isocitrate dehydrogenase (NADP); n, E1 subunit-like 2-oxoglutarate dehydrogenase, o, ATP-citrate lyase/succinyl-CoA ligase. Three squares from left to right represent WS10 vs. WW10, WS15 vs. WW15, and WS20 vs. WW20, respectively. The red squares represent significant upregulation at the 0.05 probability level. The green squares represent significant low-regulation at the 0.05 probability level. The white squares represent no significance. The red arrows show the damaged part of the chloroplast membrane in the pod wall under WS. Define the Dotted black arrow if possible

**Table 1 ijms-21-04457-t001:** Effect of drought stress on the concentration of chlorophyll in the pod wall.

Days after Pollination	Chlorophyll a (mg g^−1^)		Chlorophyll b (mg g^−1^)		Total Chlorophyll (mg g^−1^)	
	WW	WS	WW	WS	WW	WS
DAP5	0.314 ^aA^	0.141 ^aB^	0.137 ^aA^	0.055 ^aB^	0.451 ^aA^	0.196 ^aB^
DAP10	0.138 ^bA^	0.085 ^abB^	0.064 ^bA^	0.044 ^abA^	0.202 ^bA^	0.128 ^bB^
DAP15	0.094 ^bA^	0.041 ^bcB^	0.055 ^bA^	0.025 ^bcB^	0.149 ^bA^	0.066 ^cB^
DAP20	0.024 ^cA^	0.012 ^cB^	0.019 ^cA^	0.010 ^cB^	0.043 ^cA^	0.022 ^dB^

WW, well-watered; WS, water-stressed. Different small letters in the same column and different capitals in the same row meant a significant difference at the 0.05 probability level.

**Table 2 ijms-21-04457-t002:** Different expression of proteins involved in some pathways in the pod wall under drought stress.

KEGG	Accession	Proteins	Fold		
			WS10 vs. WW10	WS15 vs. WW15	WS20 vs. WW20
Amino sugar and nucleotide sugar metabolism	A0A072UKS2	PfkB family carbohydrate kinase	NS	NS	0.4
	A0A072VQZ5	UDP-D-apiose/UDP-D-xylose synthase	NS	NS	0.2
	G7JUS9	UDP-glucuronic acid decarboxylase	NS	3.3	0.4
	G7ID31	Chitinase	NS	NS	5.0
	G7LA76	Chitinase (Class Ib)/Hevein	NS	NS	3.3
Ascorbate and aldarate metabolism	A0A072TLF4	Myo-inositol oxygenase	0.4	NS	NS
	A0A072U2G7	NAD-dependent aldehyde dehydrogenase family protein	0.2	3.3	NS
	A0A072UQP6	UDP-glucose 6-dehydrogenase	2.5	NS	NS
	G7L571	UDP-glucose 6-dehydrogenase	NS	2.5	NS
	A0A072V120	UTP-glucose-1-phosphate uridylyltransferase	NS	2.5	NS
	A0A072V151	L-ascorbate oxidase	5.0	NS	NS
	A0A072VNM9	GME GDP-D-mannose-3, 5-epimerase	NS	3.3	0.1
	G7L1 × 0	GME GDP-D-mannose-3, 5-epimerase	2.5	NS	0.3
	G7JTZ5	Aldo/keto reductase family oxidoreductase	0.5	NS	NS
	G7KAG7	Thylakoid lumenal 29 kDa protein	0.3	NS	NS
beta-Alanine metabolism	A0A072UCM6	Glutamate decarboxylase	NS	NS	0.2
Carbon metabolism	G7IT85	Phosphoglycerate kinase	NS	NS	3.3
	G7IT86	Phosphoglycerate kinase	NS	NS	3.3
	G7KJZ8	Glucose-6-phosphate isomerase	NS	NS	0.4
	G7L1U4	Ribose-5-phosphate isomerase A	NS	NS	0.4
	I3S3S0	Cytosolic triosephosphate isomerase	NS	NS	2.0
	A0A072VS77	Methylenetetrahydrofolate reductase	NS	3.3	0.3
Starch and sucrose metabolism	A0A072UCM8	Phosphotransferase	0.3	NS	NS
	A0A072UKS2	PfkB family carbohydrate kinase	NS	NS	0.4
	A0A072UU47	Glycoside hydrolase family 1 protein	3.3	NS	NS
	A0A072VLQ9	Starch synthase	2.5	NS	NS
	G7IJV7	Glycoside hydrolase family 3 protein	0.3	NS	NS
	G7KJZ8	Glucose-6-phosphate isomerase	NS	NS	0.4
Citrate cycle (TCA cycle)	G7KVS0	E1 subunit-like 2-oxoglutarate dehydrogenase	NS	NS	0.3
	G7JYQ8	Aconitate hydratase	NS	NS	0.3
	B7FJJ4	Pyruvate dehydrogenase E1 beta subunit	NS	2.0	2.5
	G7KHI5	Isocitrate dehydrogenase [NADP]	NS	NS	0.2
	A2Q2V1	ATP-citrate lyase/succinyl-CoA ligase	NS	2.5	0.1
Glycine, serine and threonine metabolism	A0A072URB1	Amine oxidase	NS	NS	5.0
	A0A072V290	Amine oxidase	NS	NS	3.3
	G7J7B0	Amine oxidase	NS	NS	5.0
	A9YWS0	Serine hydroxymethyltransferase	NS	NS	0.2
	G7I9Z0	Glycine dehydrogenase [decarboxylating] protein	NS	NS	0.4
	G7JJ96	Aminomethyltransferase	NS	NS	0.3
	G7JNS2	NAD-dependent aldehyde dehydrogenase family protein	NS	NS	0.4
	G7L9H1	Phosphoserine aminotransferase	NS	NS	0.3
Linoleic acid metabolism	A0A072UMH4	Lipoxygenase	0.5	NS	NS
	G7J629	Lipoxygenase	0.4	NS	NS
	G7LIX7	Lipoxygenase	5.0	NS	NS
	G7LIY0	Lipoxygenase	5.0	NS	0.4
	G7LIY2	Lipoxygenase	10.0	NS	NS
	G7J632	Lipoxygenase	2.5	2.5	NS
Oxidative phosphorylation	A0A072URM9	Archaeal/vacuolar-type H+-ATPase subunit A	NS	2.0	NS
	A0A072V4G2	F0F1 ATP synthase subunit gamma	NS	0.5	NS
	A0A072W1H5	ATP synthase subunit beta	NS	2.5	NS
	A0A126TGR5	ATP synthase subunit alpha	NS	2.0	NS
	B7FN64	NADH dehydrogenase	NS	0.4	NS
	G7JIL4	V-type proton ATPase subunit a	NS	2.5	0.3
	G7I9M9	ATP synthase D chain	NS	5.0	NS
Phagosome	A0A072VSL4	Archaeal/vacuolar-type H+-ATPase subunit B	NS	NS	0.4
	B7FMK2	Archaeal/vacuolar-type H+-ATPase subunit E	NS	NS	3.3
	G7KSI7	Archaeal/vacuolar-type H+-ATPase subunit B	NS	NS	0.3
	G7LIN7	Tubulin beta-1 chain	NS	NS	5.0
Plant–pathogen interaction	B7FNA2	EF hand calcium-binding family protein	NS	NS	2.5
	G7I7Q4	Heat shock protein 81-2	NS	2.5	0.2
	G7IDZ4	Heat shock protein 81-2	NS	NS	0.1
	A0A072U9J1	Heat shock protein 81-2	2.5	5.0	0.3
Proteasome	A0A072TQB8	Glyceraldehyde-3-phosphate dehydrogenase	NS	NS	10
	B7FGZ8	Proteasome subunit beta type	NS	NS	2.5
	G7JTX3	6S proteasome regulatory subunit S2 1B	NS	5.0	0.3
	I3RZQ6	Proteasome subunit alpha type	NS	NS	3.3
	I3SSX1	Proteasome subunit alpha type	NS	NS	2.5
alpha-linolenic acid metabolism	G7J5N1	Uncharacterized protein	3.3	NS	NS
	Q711Q9	Allene oxide cyclase	2.5	0.1	NS

Fold change over 2.0 means significant (*p* < 0.05) upregulation and below 0.5 means significant (*p* < 0.05) downregulation. NS, nonsignificant.

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
