# Peer review of "Ultrastructural and Photosynthetic Responses of Pod Walls in Alfalfa to Drought Stress"

_ijms, 2020, doi:10.3390/ijms21124457_

Round 1
Reviewer 1 Report
The paper "Ultrastructural and photosynthetic responses of pod wall in alfalfa to drought stress" by Hui et al. (ISSN 1422-0067) is an interesting work that analyzes the photosynthetic ability of alfalfa pod walls under two different water regimes. It provides meaningful insights into the structure, chlorophyll content, specific enzymatic activities and proteomic content in the pod walls. Despite its interest, there are a few issues that prevent me from recommending its publication in its present form. I will outline some of those issues:
- First of all, English language and grammar must be thoroughly improved. Present tense and past tense are continuously mistaken, making the reading of the article very difficult. It is not clear whether the authors are talking about their findings or about other people findings. Furthermore, there are several expressions that do not make sense or are simply wrong. The article should be checked by a native English language speaker or a professional language service.
- Materials and methods:
- How was the Well Watered treatment established? Is not that amount of water an excessive treatment? The rationale behind both treatments should be explained in more detail.
- Line 111 and 112: Do you mean absorbance? Which company does the UH5300 belong to?
- In the enzymatic paragraph, there are several abbreviations that are not explained. It the authors are using a method previously applied by other authors, they should reference it. Indeed, there are a few abbreviations throughout the manuscript that are not explained. They should be explained the first time they are mentioned and used consistently afterwards.
- Line 137: "induced". Do you mean reduced?
- Line 141: concerning Solution A, Is it prepared in water?
- Line 147: Why did the authors choose the 300-4000 Da range?
- Line 151: Authors used an LSD test. Does this mean they developed an ANOVA analysis?
- Results: Supplementary data are impossible to follow, there is no way information can be checked. They should be ordered into a table format. In Table 2, it is not clear what proteins belong to each KEGG category. Similarly, legend in Fig. 4 is also confusing.
- Scientific findings and other considerations:
- Line 34: I do not think there is such thing as "resistance reaction". There are resistance mechanisms in plants, but not a unique resistance reaction.
- Line 43: Reference 15 is not about cotton.
- Line 63: "The key photosynthetic enzyme, PEPC...". It might be one of the key enzymes, but certainly not the key enzyme.
- Lines 215-216: I do not think that the data gathered by the authors allows them to say that the synthesis was inhibited, as they only analyze the presence of those proteins.
- Lines 222 and 224: "...among different duration after pollination" This expression should be corrected.
- Picture scales in Fig. 1 and 2 are almost illegible.
- Lines 299-301: authors say one thing and the contrary in just two lines. This should be explained clearly.
- Lines 372-373: This sentence does not make sense. How does the finding in the WW treatment bring you to the conclusion that follows?
- Lines 380-382: this is too redundant. Should be rewritten.
- Lines 401-402. authors are detecting the presence of chitinase, not analyzing its activity.
In conclusion, several changes should be made to the draft, particularly related to English language and clarity of some results. Besides, several statements sohuld be corrected. I encourage the authors to make such changes in order to get their significant results published in this journal.
Author Response
- First of all, English language and grammar must be thoroughly improved. Present tense and past tense are continuously mistaken, making the reading of the article very difficult. It is not clear whether the authors are talking about their findings or about other people findings. Furthermore, there are several expressions that do not make sense or are simply wrong. The article should be checked by a native English language speaker or a professional language service.
Response:
Thanks a lot for your suggestions. We have checked the tense the sentences in the manuscript and revised the mistaken parts. We have revised or rewritten the parts that mentioned and then responded your comments one by one. Please check them written in red in the manuscript.
- Materials and methods:
- How was the Well Watered treatment established? Is not that amount of water an excessive treatment? The rationale behind both treatments should be explained in more detail.
Response:
Thanks for your suggestions. We set the water treatments according to the proportion of the maximum field capacity of the soil. The water content of Well Watered treatment is 58.5%, which is around 75% of the maximum field capacity of the soil. The water content of Water Stressed treatment is 32%, which is around 25% of the maximum field capacity of the soil.
- Line 111 and 112: Do you mean absorbance? Which company does the UH5300 belong to?
Response:
Thanks for your suggestions. Yes, it is “absorbance”. We measured the absorbance by using the ultraviolet spectrophotometer. The equipment was made by Hitachi. We have added the company information in Line 122 Page 3 in the manuscript.
- In the enzymatic paragraph, there are several abbreviations that are not explained. It the authors are using a method previously applied by other authors, they should reference it. Indeed, there are a few abbreviations throughout the manuscript that are not explained. They should be explained the first time they are mentioned and used consistently afterwards.
Response:
Thanks for your suggestions. PEPC, RuBisCo, NAD-ME, NADP-ME, NADP-MDH were explained in Line 64- Line 67 Page 2. We have cited the reference for measuring the activities of enzymes in Line 125 and Line 127 Page 3.
- Line 137: "induced". Do you mean reduced?
Response:
Thanks for your suggestions. It is “reduced”. We have replaced it in Line 140 Page 3.
- Line 141: concerning Solution A, Is it prepared in water?
Response:
Yes, solution A was prepared in water.
- Line 147: Why did the authors choose the 300-4000 Da range?
Response:
Thanks for your questions. Usually this range will be set to carry out the mass spectrometry.
- Line 151: Authors used an LSD test. Does this mean they developed an ANOVA analysis?
Response:
Yes, the ANOVA analysis was used to analyze the changes of the chlorophyll concentration and the enzyme activities.
- Results: Supplementary data are impossible to follow, there is no way information can be checked. They should be ordered into a table format. In Table 2, it is not clear what proteins belong to each KEGG category. Similarly, legend in Fig. 4 is also confusing.
Response:
We have revised the supplementary files into the table format in word. We have added the blank line between the adjacent KEGG categories in the Table 2. We have revised the legend in Figure 4.
- Scientific findings and other considerations:
- Line 34: I do not think there is such thing as "resistance reaction". There are resistance mechanisms in plants, but not a unique resistance reaction.
Response:
Thanks for your suggestions. We have revised "resistance reaction" in Line 34 Page 1.
- Line 43: Reference 15 is not about cotton.
Response:
Thanks. We have revised to “castor” in Line 43 Page 1.
- Line 63: "The key photosynthetic enzyme, PEPC...". It might be one of the key enzymes, but certainly not the key enzyme.
Response:
Thanks for your suggestions. We have revised according to the suggestion in Line 63 Page 2.
- Lines 215-216: I do not think that the data gathered by the authors allows them to say that the synthesis was inhibited, as they only analyze the presence of those proteins.
Response:
The chlorophyll a-b binding proteins including I3SIG9, G7INT9, B7FIZ5, I3SZG9 were down-regulated under drought stress. Thus, we proposed that the synthesis of chlorophyll a-b binding proteins was inhibited under drought stress.
- Lines 222 and 224: "...among different duration after pollination" This expression should be corrected.
Response:
Thanks for your suggestions. We have revised the sentence in Line 225 Page 5.
- Picture scales in Fig. 1 and 2 are almost illegible.
Response:
Thanks for your suggestion. The dpi (dots per inch) of Figure 1 and Figure 2 is 300 dpi, which is suitable for publish.
- Lines 299-301: authors say one thing and the contrary in just two lines. This should be explained clearly.
Response:
We have revised the sentence. “However” is not suitable, and we revised into “in addition” in Line 303 Page 9. It was explained in Line 304-307 Page 9-10.
- Lines 372-373: This sentence does not make sense. How does the finding in the WW treatment bring you to the conclusion that follows?
Response:
Thanks for your suggestions. We have revised the sentence in Line 374 Page 12.
- Lines 380-382: this is too redundant. Should be rewritten.
Response:
Thanks for your suggestions. We have rewritten the sentence in Line 382-383 Page 12.
- Lines 401-402. authors are detecting the presence of chitinase, not analyzing its activity.
Response:
Thanks for your suggestions. The chitinase was expressed in the pod wall of alfalfa and its change under drought stress was analyzed in Line 404-405 Page 13.
In conclusion, several changes should be made to the draft, particularly related to English language and clarity of some results. Besides, several statements sohuld be corrected. I encourage the authors to make such changes in order to get their significant results published in this journal.
Reviewer 2 Report
The paper has an interesting subject and presents the ultrastructural, physiological and proteomic changes in the alfalfa pod wall under drought stress. The work is well designed, and the results are well commented. However, in the beginning, I have found difficulties in correlate WS with G and WW with Z samples. I am aware that Authors have clearly explained it, but I was wondering if there might be a way to make it easier to understand to the readers. As an example, samples subjected to proteomic analysis might be labelled with WS-G10 or only WS-10. This is just a suggestion.
-Why did the Author comment on the following results? Line 352-“In addition, drought stress induced the up-regulation of PEPC (p=0.069, fold change=1.8) in the pod wall on DAP10 line.”
According to their statement, the fold change to be significative should be over 2 and p<0.05.
-Due to the huge set of data analyzed, I suggest the Authors uploading the raw files and the Search engine outputs file on Data are available via ProteomeXchange (http://www.proteomexchange.org) to make all the information publicly available.
-Please reduce the size of the font in Table 2 because it is hard to read.
-Please detail the adopted parameters for protein identification (scores, no. of missed cleavages, etc.) the date of Medicago database downloading and the number of accessions.
-Line 137: “60 μg protein samples were induced”. Induced?
Author Response
The paper has an interesting subject and presents the ultrastructural, physiological and proteomic changes in the alfalfa pod wall under drought s
